# Learning-Augmented Dynamic Power Management with Multiple States via New Ski Rental Bounds

**Antonios Antoniadis**
University of Twente
Enschede, The Netherlands
a.antoniadis@utwente.nl

**Christian Coester**
Tel Aviv University
Tel Aviv, Israel
christian.coester@gmail.com

**Marek Eliáš**
Università Bocconi
Milan, Italy
marek.elias@unibocconi.it

**Adam Polak**
EPFL
Lausanne, Switzerland
adam.polak@epfl.ch

**Bertrand Simon**
IN2P3 Computing Center, CNRS
Villeurbanne, France
bertrand.simon@cnrs.fr

## Abstract

We study the online problem of minimizing power consumption in systems with multiple power-saving states. During idle periods of unknown lengths, an algorithm has to choose between power-saving states of different energy consumption and wake-up costs. We develop a learning-augmented online algorithm that makes decisions based on (potentially inaccurate) predicted lengths of the idle periods. The algorithm's performance is near-optimal when predictions are accurate and degrades gracefully with increasing prediction error, with a worst-case guarantee almost identical to the optimal classical online algorithm for the problem. A key ingredient in our approach is a new algorithm for the online ski rental problem in the learning augmented setting with tight dependence on the prediction error. We support our theoretical findings with experiments.

## 1 Introduction

Energy represents up to 70% of total operating costs of modern data centers [41] and is one of the major quality-of-service parameters in battery-operated devices. In order to ameliorate this, contemporary CPUs are equipped with sleep states to which the processor can transition during periods of inactivity. In particular, the ACPI-standard [25] specifies that each processor should possess, along with the active state $C0$ that is used for processing tasks, at least one sleep state $C1$. Modern processors generally possess more sleep states $C2, \ldots$; for example, current Intel CPUs implement at least 4 such $C$-states [19]. Apart from CPUs, such sleep states appear in many systems ranging from hard drives or mobile devices to the start-stop feature found in many cars, and are furthermore often employed when rightsizing data centers [2].

Intuitively, in a "deeper" sleep state, the set of switched-off components will be a superset of the corresponding set in a more shallow sleep state. This implies that the *running cost* for residing in that deeper state will be lower, but the *wake-up* cost to return to the active state $C0$ will be higher compared to a more shallow sleep state. In other words, there is a tradeoff between the running and the wake-up cost. During each idle period, a *dynamic power management (DPM)* strategy has to decide in which state the system resides at each point in time, without a-priori knowledge about the duration of the idle period. Optimally managing these sleep states is a challenging problem due to its online nature. On the one hand, transitioning the system to a too deep state could be highly suboptimal if the idle period ends shortly after. On the other hand, spending too much idle time in a shallow state would accumulate high running costs. The impact of DPM strategies in practice

35th Conference on Neural Information Processing Systems (NeurIPS 2021).

has been studied for instance in data centers, where each machine may be put to a sleep mode if no request is expected. See the study of Lim et al. [34] on multi-tier data centers.

The special case of 2-state DPM systems, i.e., when there is only a single sleep state (besides the active state), is essentially equivalent to the *ski rental* problem, one of the most classical problems and of central importance in the area of online optimization [39; 26]. This problem is defined as follows: A person goes skiing for an unknown number of days. On every day of skiing, the person must decide whether to continue renting skis for one more day or to buy skis. Once skis are bought there will be no more cost on the following days, but the cost of buying is much higher than the cost of renting for a day. It is easy to see that this captures a single idle period of DPM with a single sleep state whose running cost is 0: The rental cost corresponds to the running cost of the active state and the cost of buying skis corresponds to the wake-up cost; transitioning to the sleep state corresponds to buying skis. Given this equivalence, the known 2-competitive deterministic algorithm and $e/(e-1) \approx 1.58$-competitive randomized algorithm for ski rental carry over to 2-state DPM, and these competitive ratios are tight. In fact, it was shown by Irani et al. [28] and Lotker et al. [36] that the same competitive ratios carry over even to multi-state DPM. Ski rental, also known as *rent-or-buy* problem, is a fundamental problem appearing in many domains not restricted to computer hardware questions. For the AI community, this problem for example implicitly appears in expert learning with switching costs: paying the price to switch to a better expert allows to save expenses in the future.

Beyond these results for the classical online setting, [28] also gave a deterministic $e/(e-1)$-competitive algorithm for the case in which the length of the idle periods is repeatedly drawn from a fixed, and known, probability distribution. When the probability distribution is fixed but unknown they developed an algorithm that *learns* the distribution over time and showed that it performs well in practice. Although it is perhaps not always reasonable to assume a fixed underlying probability distribution for the length of idle periods, real-life systems do often follow periodical patterns so that these lengths can indeed be frequently predicted with adequate accuracy, see Chung et al. [18] for a specific example. Nevertheless, it is not hard to see that blindly following such predictions can lead to arbitrarily bad performance when predictions are faulty. The field of *learning-augmented* algorithms [38] is concerned with algorithms that incorporate predictions in a robust way.

In this work, we introduce multi-state DPM to the learning-augmented setting. Extending ideas of [28] and [36], we give a reduction from multi-state DPM to ski rental that is applicable to the learning-augmented setting. Although ski rental has been investigated through the learning-augmented algorithms lens before [40; 44], earlier work has focused on the optimal trade-off between *consistency* (i.e., the performance when predictions are accurate) and *robustness* (i.e., the worst-case performance). To apply our reduction from DPM to ski rental, we require more refined guarantees for learning-augmented ski rental. To this end we develop a new learning-augmented algorithm for ski rental that obtains the optimal trade-off between consistency and dependence on the prediction error. Our resulting algorithm for DPM achieves a competitive ratio arbitrarily close to 1 in case of perfect predictions and its performance degrades gracefully to a competitive ratio arbitrarily close to the optimal robustness of $e/(e-1) \approx 1.58$ as the prediction error increases.

**Potential negative societal impact.** This is a work of theoretical nature and we are not aware of potential negative societal impact. That said, we cannot rule out future misuse of the contained theoretical knowledge.

## 1.1 Formal definitions

**Problem definition.** In the problem of *dynamic power management (DPM)*, we are given $k + 1$ power states denoted by $0, 1, \ldots, k$, with power consumptions $\alpha_0 > \cdots > \alpha_k \geq 0$ and wake-up costs $\beta_0 < \cdots < \beta_k$. For state 0, we have $\beta_0 = 0$ and we call this the *active state*. The input is a series of *idle periods* of lengths $\ell_1, \ldots, \ell_T$ received online, i.e., the algorithm does not know the length of the current period before it ends. During each period, the algorithm can transition to states with lower and lower power consumption, paying energy cost $x\alpha_i$ for residing in state $i$ for time $x$. If $j$ is the state at the end of the idle period, then it has to pay the wake-up cost $\beta_j$ to transition back to the active state 0. The goal is to minimize the total cost.

In the *learning-augmented* setting, the algorithm receives at the beginning of the $i$th idle period a prediction $\tau_i \geq 0$ for the value of $\ell_i$ as additional input. We define $\eta_i := \alpha_0|\tau_i - \ell_i|$ to be the error of the $i$th prediction, and $\eta := \sum_i^T \eta_i$ to be the total prediction error.

(Continuous-time) *ski rental* is the special case of DPM with $k = 1$, $\alpha_1 = 0$ and a single idle period of some length $\ell$. In this case, we call $\alpha := \alpha_0$ the rental cost, $\beta := \beta_1$ the buying cost, and $\ell$ the length of the ski season. In learning-augmented ski rental, we write the single prediction as $\tau := \tau_1$.

**$(\rho, \mu)$-competitiveness.** Classical online algorithms are typically analyzed in terms of *competitive ratio*. A (randomized) algorithm $\mathcal{A}$ for an online minimization problem is said to be $\rho$-*competitive* (or alternatively, obtain a *competitive ratio* of $\rho$) if for any input instance,

$$cost(\mathcal{A}) \leq \rho \cdot \text{OPT} + c, \tag{1}$$

where $cost(\mathcal{A})$ and OPT denote the (expected) cost of $\mathcal{A}$ and the optimal cost of the instance and $c$ is a constant independent of the *online* part of the input (i.e., the lengths $\ell_i$ in case of DPM). For the ski rental problem one requires $c = 0$, since the trivial algorithm that buys at time 0 has constant cost $\beta$.

In the learning-augmented setting, for $\rho \geq 1$ and $\mu \geq 0$, we say that $\mathcal{A}$ is $(\rho, \mu)$-*competitive* if

$$cost(\mathcal{A}) \leq \rho \cdot \text{OPT} + \mu \cdot \eta \tag{2}$$

for any instance, where $\eta$ is the prediction error. This corresponds to a competitive ratio of $\rho + \mu \frac{\eta}{\text{OPT}}$ (with $c = 0$). While this could be unbounded as $\eta/OPT \to \infty$, our DPM algorithm achieves a favorable competitive ratio even in this case (see Theorem 5, where we take the minimum over a range of pairs $(\rho, \mu)$, including $\mu = 0$).

For a $(\rho, \mu)$-competitive algorithm, $\rho$ is also called the *consistency* (i.e., competitive ratio in case of perfect predictions) while $\mu$ describes the dependence on the prediction error.

## 1.2 Our results

Our first result is a $(\rho, \mu)$-*competitive algorithm* for ski rental that achieves the optimal $\mu$ corresponding to the given $\rho$. For $\rho \in [1, \frac{e}{e-1}]$, let

$$\mu(\rho) := \max\left\{\frac{1 - \rho\frac{e-1}{e}}{\ln 2}, \rho(1-T)e^{-T}\right\}, \tag{3}$$

where $T \in [0, 1]$ is the solution to $T^2 e^{-T} = 1 - \frac{1}{\rho}$. Let $\tilde{\rho} \approx 1.16$ be the value of $\rho$ for which both terms in the maximum yield the same value. The first term dominates for $\rho > \tilde{\rho}$ and the second term if $\rho < \tilde{\rho}$. Note that $\mu(1) = 1$ and $\mu\left(\frac{e}{e-1}\right) = 0$. See Figure 1 (left) for an illustration.

**Theorem 1.** *For any $\rho \in [1, \frac{e}{e-1}]$, there is a $(\rho, \mu(\rho))$-competitive randomized algorithm for learning-augmented ski rental, i.e., given a prediction with error $\eta$, its expected cost is at most $\rho \,\text{OPT} + \mu(\rho) \cdot \eta$.*

Note that $\rho < 1$ is impossible for any algorithm (due to the case $\eta = 0$) and $\rho > \frac{e}{e-1}$ is uninteresting since $\rho = \frac{e}{e-1}$ already achieves the best possible value of $\mu = 0$.

We also prove a lower bound showing that $\mu(\rho)$ defined in (3) is the best possible.

**Theorem 2.** *For any $\rho \in [1, \frac{e}{e-1}]$ and any (randomized) algorithm $\mathcal{A}$, there is a ski rental instance with some prediction error $\eta$ such that the expected cost of $\mathcal{A}$ is at least $\rho \,\text{OPT} + \mu(\rho)\eta$.*

However, for most values of the prediction $\tau$ it is possible to achieve a better $\mu < \mu(\rho)$, and $\mu(\rho)$ only captures the worst case over all possible predictions $\tau$. The proof of Theorem 1 is sketched in Section 2. The complete proofs of Theorems 1 and 2 are provided in the full version [8] in the supplementary material.

In Section 3, we give a reduction from DPM to ski rental in the learning-augmented setting, provided that the ski rental algorithm satisfies a natural monotonicity property (defined formally in Section 3):

**Lemma 3.** *If there is a monotone $(\rho, \mu)$-competitive ski rental algorithm, then there is a $(\rho, \mu)$-competitive algorithm for DPM.*

Since our ski rental algorithm is monotone, this directly yields a $(\rho, \mu(\rho))$-competitive algorithm for DPM. From the special case $(\rho, \mu) = \left(\frac{e}{e-1}, 0\right)$, this theorem directly implies the following result for classical DPM (without predictions), which was first proved by Lotker et al. [36] for the equivalent multi-slope ski rental problem:

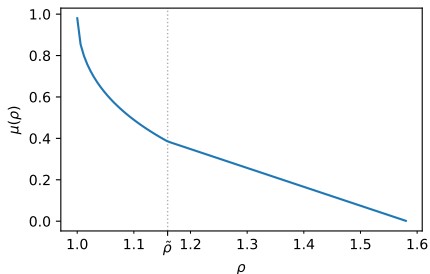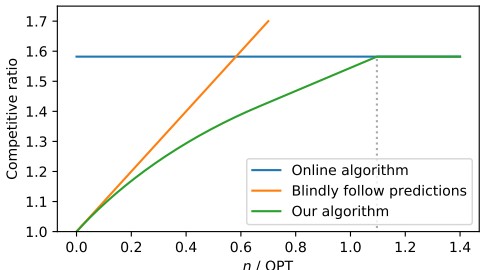

Figure 1: Illustration of $\mu(\rho)$ and of the resulting competitive ratio in function of $\eta/\mathrm{OPT}$.

**Corollary 4** ([36]). *There is a $\frac{e}{e-1}$-competitive randomized online algorithm for DPM (without predictions).*

Using techniques from online learning, in a way similar to [5], we show in Section 4 how to achieve "almost" $(\rho, \mu(\rho))$-competitiveness simultaneously for *all* $\rho$:

**Theorem 5.** *For any $\epsilon > 0$, there is a learning-augmented algorithm $\mathcal{A}$ for dynamic power management whose expected cost can be bounded as*

$$cost(\mathcal{A}) \leq (1+\epsilon) \min \left\{ \rho \, \mathrm{OPT} + \mu(\rho) \cdot \eta \,\middle|\, \rho \in [1, \tfrac{e}{e-1}] \right\} + O\!\left(\tfrac{\beta_k}{\epsilon} \log \tfrac{1}{\epsilon}\right).$$

The above theorem gives a competitive ratio arbitrarily close to $\min\{\rho + \mu(\rho) \cdot \frac{\eta}{\mathrm{OPT}}\}$, which is equal to 1 if $\eta = 0$ and never greater than $\frac{e}{e-1}$. In particular, we achieve a performance that degrades gracefully from near-optimal consistency to near-optimal robustness as the error increases.[1] See Figure 1 (right) for an illustration.

In Section 5, we illustrate the performance of these algorithms by simulations on synthetic datasets, where the dependence on the prediction error can be observed as expected from theoretical results.

## 1.3 Related work

**Learning-Augmented Algorithms.** Learning augmented algorithms have been a very active area of research since the seminal paper of Lykouris and Vassilvitskii [37]. We direct the interested reader to a survey [38] by Mitzenmacher and Vassilvitskii, as well as [7; 20; 37; 5; 42; 43; 35; 32] for recent results on secretary problems, paging, $k$-server as well as scheduling problems. In the following we survey some results in the area more closely related to our work.

The ski rental problem has already been studied within the context of learning augmented algorithms. Here, the main objective was to optimize the tradeoff between consistency and robustness (performance on perfect predictions and worst-case performance). The first results are due to Purohit et al. [40] who propose a deterministic and a randomized algorithm. A hyperparameter allows to choose a prescribed consistency $\rho$ and leads to a corresponding robustness. They also present a linear dependency on the error: their randomized algorithm is $(\rho, \rho)$-competitive for $\rho \geq 1$, with larger $\rho$ allowing for better robustness. Note that such a guarantee of $(\rho, \rho)$-competitiveness is not valuable in our model where we do not focus on robustness as blindly following the predictions leads to a $(1, 1)$-competitive algorithm. Wei and Zhang [44] show that the consistency / robustness tradeoff achieved by the randomized algorithm of [40] is Pareto-optimal. Angelopoulos et al. [4] propose a deterministic algorithm achieving a Pareto-optimal consistency / robustness tradeoff, but with no additional guarantee when the error is small. Interestingly, these algorithms not focusing on $(\rho, \mu)$-competitiveness are naturally monotone, so easily extend to DPM by Lemma 3, contrarily to the tight algorithm we present in this paper. Nevertheless, experimental data (Section 5) seem to indicate that our algorithm optimizing $(\rho, \mu)$-competitiveness for ski rental leads to better algorithms for DPM. A variant with multiple predictions was also studied in [21].

As we will see in Section 4, DPM can be cast as a problem from the class of *Metrical Task Systems (MTS)*. Antoniadis et al. [5] gave a learning-augmented algorithm for MTS that can be interpreted as $(1, 4)$-competitive within their prediction setup.

---

[1] At first glance, our consistency and robustness might seem to contradict the lower bound of Wei and Zhang [44] for ski rental. However, [44] crucially uses $c = 0$ in the definition of competitiveness for ski rental.

A different problem related to energy conservation is the classical online speed scaling problem, which was recently studied in the learning-augmented setting by Bamas et al. [11].

**DPM.** The equivalence between 2-state DPM and ski rental is mentioned in [39]. Therefore the well-known 2-competitive deterministic and an $e/(e-1)$-competitive randomized algorithm [30] for the classical ski rental problem carry over to 2-state DPM, and these bounds are known to be tight.

Irani et al. [28] present an extension of the 2-competitive algorithm for two-state DPM to multi-state DPM that also achieves a competitive ratio of 2. Furthermore they give an $e/(e-1)$-competitive algorithm for the case that the lengths of the idle periods come from a fixed probability distribution.

Lotker et al. [36] consider what they call *multi-slope ski rental* which is equivalent to the DPM problem. Among other results, they show how to reduce a $(k+1)$-slope ski rental instance to $k$ classical ski rental instances. The reduction from DPM to ski rental presented in this paper is similar, but more general in order to also be applicable in the presence of predictions with the introduced $(\rho, \mu)$-competitiveness. They furthermore show how to compute the best possible randomized strategy for any instance of the problem.

There have been several previous approaches that try to predict the length of an idle interval (see, e.g., [18; 28], and the survey of Benini et al. [13]). However, the proposed approaches to use these predictions are not robust against a potentially high prediction error.

Augustine et al. [10] investigate a problem generalizing DPM where transition cost is paid for going to a deeper sleep state rather than waking up and these transition costs may be non-additive (i.e., it can be cheaper to skip states). Albers [2] studies the offline version of the problem with multiple, parallel devices and shows that it can be solved in polynomial time.

Irani et al. [29] introduced a 2-state problem where jobs that need to be processed have a release-time, a deadline and a required processing time. This gives further flexibility to the system to schedule the jobs and create periods of inactivity so as to maximize the energy-savings by transitioning to the sleep state. For the offline version, there is an exact polynomial-time algorithm due to Baptiste et al. [12]. Recently, a 3-approximation algorithm for the multiprocessor-case was developed [6].

Another related problem consists of deciding which components of a data-center should be powered on or off in order to process the current load on the set of active components (see, e.g., [3]). A similar problem, where jobs have individual processing times for each machine, was studied in [31; 33]. Helmbold et al. [23] considered the problem of spinning down the disk of a mobile computer when idle times are expected, which is another instance of DPM.

Several surveys cover DPM, see for example [13; 1; 27].

## 2 New algorithm for ski rental

For $\rho \in [1, \frac{e}{e-1}]$, we present a $(\rho, \mu(\rho))$-competitive algorithm for (learning augmented) ski rental, proving Theorem 1. The next lemma shows that it suffices to give such an algorithm for $\alpha = \beta = 1$.

**Lemma 6.** *An algorithm $\mathcal{A}'$ that is $(\rho, \mu)$-competitive for instances of the ski rental problem with $\alpha = \beta = 1$ implies a $(\rho, \mu)$-competitive algorithm $\mathcal{A}$ for arbitrary $\alpha, \beta > 0$.*

*Proof idea.* Simulate $\mathcal{A}$' with prediction $\frac{\alpha}{\beta}\tau$. If it buys at time $t'$, then $\mathcal{A}$ buys at time $t = \frac{\beta}{\alpha}t'$. □

A key difference between proving $\rho$-competitiveness in the classical online setting and $(\rho, \mu)$-competitiveness in the learning-augmented setting is the following. In the online setting without predictions, a greedy algorithm that buys with the highest affordable probability is optimal. This relies on the fact that the right-hand side of (1) is a monotone and concave function of the skiing season length. In contrast, the right-hand side of (2) is neither monotone (if $\tau > 1$) nor concave (regardless of $\tau$), which complicates the description and especially the analysis of our algorithm.

### 2.1 Description of the algorithm

We next describe our randomized algorithm for instances with $\alpha = \beta = 1$, which can then be used to solve arbitrary ski rental instances using Lemma 6. Our algorithm is fully specified by the cumulative

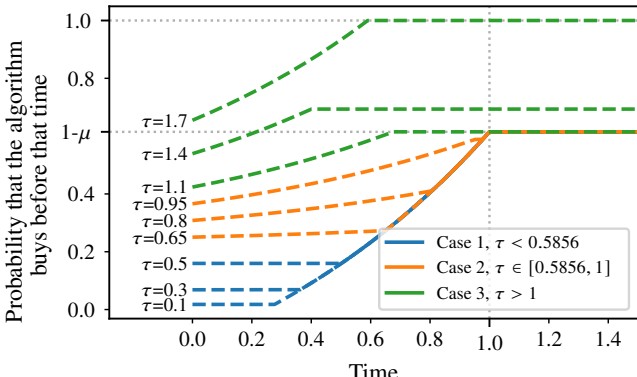

Figure 2: Our $(\rho, \mu)$-competitive ski rental algorithm for $\rho = \tilde{\rho} \approx 1.1596$ and $\mu = \mu(\tilde{\rho}) \approx 0.3852$. The figure presents the cumulative distribution functions of the time of buying for several prediction values $\tau$. Here $\alpha = \beta = 1$, i.e., at time $t = 1$ buying and renting has equal costs.

distribution function (CDF) $F_\tau$ of the time when the algorithm buys skis. The algorithm then draws a $p \in [0, 1]$ uniformly at random and buys at the earliest time $t \in [0, \infty)$ such that $F_\tau(t) \geq p$. The CDF $F_\tau$ will depend on the given prediction $\tau \geq 0$ as well as the fixed $\rho$ and $\mu$, which can be chosen as $\mu = \mu(\rho)$, see Equation (3).

**Definition of the CDF (see Figure 2)**  We denote by $P_0$ the probability of buying at time 0 and, for any $t > 0$, we denote by $p_t$ the probability density of buying at time $t$, so that the probability that the algorithm buys by time $x$ can be expressed as

$$F_\tau(x) = P_0 + \int_0^x p_t \, dt.$$

For convenience, we also specify the probability $P_\infty = 1 - (P_0 + \int_0^\infty p_t \, dt)$ of never buying.

To define $P_0$ and $p_t$, we distinguish three cases depending on the value of the prediction $\tau$. Note that we always have $0 \leq \mu \leq 1 \leq \rho \leq \frac{e}{e-1}$.

**Case 1: $\mu\tau < \mu - \rho + 1$.**  We choose

$$P_0 = \frac{\tau(\rho - 1)}{1 - \tau}, \qquad p_t = \begin{cases} \rho e^{t-1} & \text{for } t \in (b, 1] \\ 0 & \text{otherwise} \end{cases}, \qquad P_\infty = \min\{\mu, 1 - P_0\},$$

where $b \in [\tau, 1]$ is chosen such that $P_0 + P_\infty + \int_b^1 \rho e^{t-1} dt = 1$, in order to have the sum of probabilities equal to 1. Note that if $P_0 \geq 1 - \mu$, we have $b = 1$ and $p_t = 0$ for all $t > 0$.

**Case 2: $\mu - \rho + 1 \leq \mu\tau$ and $\tau \leq 1$.**  We choose

$$P_0 = \mu\tau, \qquad p_t = \begin{cases} (\mu\tau + \rho - \mu - 1)e^t & \text{for } t \leq a \\ \rho e^{t-1} & \text{for } t \in (b, 1] \\ 0 & \text{otherwise} \end{cases}, \qquad P_\infty = \min\{\mu, 1 - P_0\},$$

where $a \in [0, \tau]$ is chosen maximal such that $P_0 + P_\infty + \int_0^a (\mu\tau + \rho - \mu - 1)e^t dt \leq 1$, and $b \in [\tau, 1]$ is chosen so that $P_0 + P_\infty + \int_0^a (\mu\tau + \rho - \mu - 1)e^t dt + \int_b^1 \rho e^{t-1} dt = 1$ in order to have the sum of probabilities equal to 1. In case $\rho = \frac{e}{e-1}$, we have $\mu = 0$ and $(\mu\tau + \rho - \mu - 1)e^t = (\rho - 1)e^t = \rho e^{t-1}$, recovering the classical online algorithm of Karlin et al. [30].

**Case 3: $\tau > 1$.**  If $\mu\tau \geq 1$, we buy at time 0. Otherwise, we choose

$$P_0 = \mu\tau, \qquad p_t = \begin{cases} (\mu\tau + \rho - \mu - 1)e^t & \text{if } t \leq T \\ 0 & \text{if } t > T \end{cases}, \qquad P_\infty = \rho - \mu - (\mu\tau + \rho - \mu - 1)e^T,$$

where $T$ is the number closest to $\tau - 1$ that satisfies

$$e^T \leq \frac{\rho - \mu}{\mu\tau + \rho - \mu - 1} \qquad \text{(equivalently } P_\infty \geq 0) \qquad (4)$$

$$e^T \geq \frac{\rho - 2\mu}{\mu\tau + \rho - \mu - 1} \qquad \text{(equivalently } P_\infty \leq \mu). \qquad (5)$$

Thus, either $T = \tau - 1$ if this choice satisfies both bounds, or $T$ is at an endpoint of the feasible interval prescribed by (4) and (5).

## 2.2 Sketch of the analysis

Our algorithm is $(\rho, \mu)$-competitive if and only if for all $x \geq 0$ we have

$$cost(x) := P_0 + \int_0^x (1+t)\, p_t\, dt + \int_x^\infty x\, p_t\, dt + x P_\infty \leq \rho \min\{x, 1\} + \mu|\tau - x|, \qquad (6)$$

where $cost(x)$ denotes the expected cost of the algorithm in the case when $\ell = x$: If we intend to buy at some time $t$ and $t < x$, we pay $1 + t$, otherwise we pay $x$. On the right hand side, $\min\{x, 1\}$ is the optimal cost and $|\tau - x|$ is the prediction error, assuming $\alpha = \beta = 1$.

We first sketch the analysis for Case 2, and then discuss the differences in Case 1. These cases are relatively simple. Case 3 is far more involved and we will only sketch the ideas.

**Case 2:** For the algorithm to be well defined, we need to choose $\mu$ such that a suitable $b \in [\tau, 1]$ exists. For $\mu = \mu(\rho)$, this is ensured by the inequality $\mu(\rho) \geq \frac{1 - \rho\frac{e-1}{e}}{\ln 2}$ from the definition of $\mu(\rho)$, which implies existence of such $b$ for any value of $\tau$. If $\tau = \ln 2$, then $\mu = \mu(\rho)$ is in fact the smallest possible, allowing only $b = \tau$. For other values of $\tau$, suitable $b$ exists also for smaller values of $\mu$. We now show that (6) is satisfied.

Note that (6) is tight for $x = 0$, with both sides equal to $\mu\tau$. To obtain (6) for all $x > 0$, it suffices to show that the derivative of the left-hand side with respect to $x$ is at most the derivative of the right-hand side (where derivatives exist). For $x \in (0, \infty) \setminus \{a, b, 1\}$, we have

$$\frac{d}{dx} cost(x) = (1+x)p_x + \int_x^\infty p_t\, dt - x p_x + P_\infty = p_x + \int_x^\infty p_t\, dt + P_\infty.$$

For $x \in (0, a)$ this yields

$$\frac{d}{dx} cost(x) = p_x + 1 - P_0 - (p_x - p_0) = 1 - \mu\tau + (\mu\tau + \rho - \mu - 1)e^0 = \rho - \mu,$$

which is equal to the derivative of the right-hand side of (6). For $x \in (a, b)$, $\frac{d}{dx} cost(x)$ is even smaller because $p_x$ is 0, and the derivative of the right-hand side of (6) is $\rho - \mu$ or $\rho + \mu$. For $x \in (b, 1)$,

$$\frac{d}{dx} cost(x) = p_x + \int_x^\infty p_t\, dt + P_\infty = p_x + (p_1 - p_x) + P_\infty = \rho + P_\infty \leq \rho + \mu,$$

which is equal to the derivative of the right-hand side of (6). Finally, for $x > 1$ we have $\frac{d}{dx} cost(x) = P_\infty \leq \mu$ and the derivative of the right-hand side is also $\mu$.

**Case 1:** The reason we cannot define $p_t$ in the same way as in Case 2 is that $p_t$ would be negative for $t \leq a$ (i.e., the algorithm would try to sell skis that it bought at time 0, which is not allowed). We therefore choose $P_0$ such that (6) is tight for $x = \tau$ if we do not buy in the interval $(0, \tau]$. The remainder of the proof of (6) is similar to Case 2. For $\mu = \mu(\rho)$, the existence of $b \in [\tau, 1]$ follows from the inequality $\mu \geq \rho(1 - T)e^{-T}$ in the definition of $\mu(\rho)$. Note that such $\mu$ is the smallest possible for $\tau = 1 - T$.

**Case 3:** The first step in the analysis of Case 3 is to derive an inequality involving $\rho$, $\mu$, $\tau$ and $T$ that is equivalent to the algorithm being $(\rho, \mu)$-competitive. Denoting by $\mu_\tau(\rho)$ the minimal $\mu$ satisfying this inequality, it suffices to show that $\mu_\tau(\rho) \leq \mu(\rho)$ for all $\tau > 1$. The difficulty is that no closed-form expression for $\mu_\tau(\rho)$ exists. However, we are still able to show that $\tau \mapsto \mu_\tau(\rho)$ can have a local maximum only if $T = \tau - 1$, and therefore $\sup_{\tau > 1} \mu_\tau(\rho)$ is achieved either for $\tau \to 1$ or when $T = \tau - 1$. This allows us to eliminate $\tau$ from the aforementioned inequality, and we can show that $\mu = \mu(\rho)$ satisfies the remaining inequality (with tightness occurring for $\rho \leq \tilde{\rho}$ and $\tau = T + 1$).

A complete proof of Theorem 1 is in the full version of the paper in the supplementary material.

# 3 Reduction from DPM to ski rental

We now give a reduction from DPM to ski rental in the learning-augmented setting (Lemma 3), provided that the ski rental algorithm satisfies the following monotonicity property: We say that a ski rental algorithm for rental cost $\alpha = 1$ and buying cost $\beta = 1$ is *monotone* if its CDF $F_\tau$ for the buying time when given prediction $\tau$ satisfies

$$F_\tau(t) \leq F_{\tau'}(t) \qquad \text{for all } t \geq 0 \text{ and } \tau < \tau'.$$

Intuitively, this property is very natural: The longer the predicted duration of skiing, the greater should be our probability of buying. Indeed, our algorithm satisfies this property:

**Lemma 7.** *For $\mu = \mu(\rho)$, the $(\mu, \rho)$-competitive ski rental algorithm from Section 2 is monotone, i.e., its CDF $F_\tau$ when given prediction $\tau$ satisfies $F_\tau(t) \leq F_{\tau'}(t)$ for all $t \geq 0$ and $\tau < \tau'$.*

For many $\tau$ one could actually achieve a better $\mu_\tau(\rho) < \mu(\rho)$. However, somewhat surprisingly the optimal such algorithm would *not* be monotone. The monotonicity of our algorithm therefore crucially relies on our specific description (in particular the choice of $a$ and $b$), which only aims for $(\rho, \mu(\rho))$-competitiveness with $\mu(\rho) = \sup_\tau \mu_\tau(\rho)$.

Combining Theorem 1, Lemma 3 and Lemma 7, we get:

**Corollary 8.** *For every $\rho \in [1, \frac{e}{e-1}]$, there is a $(\rho, \mu(\rho))$-competitive algorithm for DPM.*

To prove Lemma 3, it suffices to describe a $(\rho, \mu)$-competitive algorithm for the special case of DPM with a single idle period: Running such an algorithm for each individual period yields a $(\rho, \mu)$-competitive algorithm for DPM with any number of idle periods, since we can simply sum inequality (2) over all periods to obtain the corresponding inequality for the entire instance.

Consider now a single idle period of length $\ell$ for DPM. We first recall some observations of Irani et al. [28] about the optimal *offline* algorithm: It is easy to see that the optimal offline algorithm would transition to some state $j$ only once at the beginning of the period and remain there throughout the period, paying cost $\alpha_j \ell + \beta_j$. Thus, state $j$ is preferred over state $j-1$ if and only if $\alpha_{j-1}\ell + \beta_{j-1} > \alpha_j \ell + \beta_j$, or equivalently $\ell > t_j := \frac{\beta_j - \beta_{j-1}}{\alpha_{j-1} - \alpha_j}$. We may assume without loss of generality that $t_1 < \cdots < t_k$: Indeed, suppose $t_{j+1} \leq t_j$, then state $j$ is redundant because whenever $j$ is preferred over $j-1$, then $j+1$ is preferred over $j$. Defining $t_0 := 0$ and $t_{k+1} := +\infty$, we get a partition $[0, +\infty) = \bigcup_{j=0}^{k} I_j$, where $I_j = [t_j, t_{j+1})$. We can then express the cost of the offline optimum as

$$\text{OPT} = \alpha_{j^*}\ell + \beta_{j^*}, \text{ with } j^* \text{ such that } \ell \in I_{j^*}. \tag{7}$$

In the online setting, we of course do not know $\ell$. The idea of our algorithm (similar to [36]) is to simulate $k$ ski rental algorithms $\mathcal{A}_1, \ldots, \mathcal{A}_k$ in parallel, where the task of $\mathcal{A}_j$ is to decide whether it is time to transition from the state $j-1$ to $j$. For this, we choose $\mathcal{A}_j$ to be an algorithm for ski rental with rental cost $\alpha_{j-1} - \alpha_j$ and buying cost $\beta_j - \beta_{j-1}$. Let $F_\tau$ be the CDF of the buying time of a *monotone* ski rental algorithm (for $\alpha = \beta = 1$) when given prediction $\tau$. Recalling our reduction from arbitrary $\alpha$ and $\beta$ to the case $\alpha = \beta = 1$ in Lemma 6, the CDF of $\mathcal{A}_j$ is given by

$$F^j(t) := F_{\tau/t_j}\left(t/t_j\right). \tag{8}$$

An outline of our algorithm is given in Algorithm 1.

---

**Algorithm 1:** DPM with a single idle period

**for** *j=1,…,k* **do**
    Let $F^j$ be as defined by (8), induced by a monotone $(\rho, \mu)$-competitive ski rental algorithm;
Choose $p \in [0, 1]$ uniformly at random;
At any time $t$: choose state $j = \max\{j : F^j(t) \geq p\}$;

---

The proof that Algorithm 1 is $(\rho, \mu)$-competitive relies on the fact that $F^j(t)$ is non-increasing in $j$:

$$F^{j-1}(t) = F_{\tau/t_{j-1}}\left(t/t_{j-1}\right) \geq F_{\tau/t_j}\left(t/t_{j-1}\right) \geq F_{\tau/t_j}\left(t/t_j\right) = F^j(t),$$

where we used $t_{j-1} < t_j$ in both inequalities, the first inequality uses monotonicity of the ski rental algorithm and the second inequality uses that any CDF is non-decreasing. Thus, algorithm $\mathcal{A}_j$ signals transitioning from state $j-1$ to $j$ no earlier than $\mathcal{A}_{j-1}$ signals transitioning from state $j-2$ to $j-1$.

**Conversion to a prudent algorithm.** It was shown in Lotker et al. [36, Theorem 4.2] that any DPM algorithm can be converted (online) into a so-called *prudent* one that assigns a non-zero probability to at most two adjacent power states. The resulting algorithm pays the same expected wake-up cost but can only have smaller running cost than the original non-prudent algorithm. Hence, any implementation should apply this conversion, which we describe in the supplementary material.

## 4 Finding the best trade-off online

Our goal is to design an algorithm whose performance almost matches that of Corollary 8 simultaneously for *all* $\rho$, proving Theorem 5. It will be useful to view DPM as a Metrical Task System.

**Metrical Task Systems (MTS).** Metrical Task Systems (MTS), introduced by Borodin et al. [15], is a broad class of online problems containing many other problems as special cases. In MTS, we are given a metric space $M$ of *states*. We start at a predefined initial state $x_0$. At each time $t = 1, 2, \ldots, T$, we are presented with a *cost function* $c_t \colon M \to \mathbb{R}_+$. Then, we have to choose our new state $x_t$ and pay $\text{dist}(x_{t-1}, x_t) + c_t(x_t)$, where $\text{dist}(x_{t-1}, x_t)$ is the distance between $x_{t-1}$ and $x_t$ in $M$. The objective is to minimize the overall cost incurred over time.

To formulate DPM as a Metrical Task System, we choose states $0, 1, \ldots, k$ corresponding to the power states, with distances $\text{dist}(i, j) = \frac{1}{2}|\beta_i - \beta_j|$, so that the cost of switching from the state 0 to $j$ and back is $\beta_j$. We choose 0 as the initial state. We discretize time in the DPM instance using time steps of some small length $\delta > 0$. At each time step belonging to some idle period, we issue a cost function $c$ such that $c(j) = \delta\alpha_j$ for each $j = 0, \ldots, k$. At the end of each idle period, we issue a cost function where $c(0) = 0$ and $c(j) = +\infty$ for $j = 1, \ldots, k$, which forces any algorithm to move back to the active state.

We use the result of Blum and Burch [14] to combine multiple instances of our algorithm with different parameters $\rho$.

**Theorem 9** (Blum and Burch [14])**.** *There is an algorithm which, given $N$ online algorithms $A_1, \ldots A_N$ for an MTS with diameter $D$ and $\epsilon_1 < 1/2$, achieves expected cost at most*

$$(1 + \epsilon_1) \cdot \min_i \{cost(A_i)\} + O(D/\epsilon_1) \ln N.$$

Using this result, the straightforward proof of Theorem 5 is given in the supplementary material. Here, we just note that we choose a suitable set $P \subset [1, \frac{e}{e-1}]$ of size $O(1/\epsilon_2)$ so that the combination of our $(\rho, \mu(\rho))$-competitive algorithms for all $\rho \in P$ using the algorithm of Blum and Burch [14] achieves expected cost at most

$$(1 + \epsilon_1)(1 + \epsilon_2) \min_{\rho \in [1, e/(e-1)]} \left\{ \rho \, \text{OPT} + \mu(\rho)\eta \right\} + O\left( \frac{\beta_k}{\epsilon_1} \cdot \ln \frac{1}{\epsilon_2} \right).$$

In the supplementary material, we also argue how using results on *shifting/dynamic* regret [17; 16; 22; 24] can be used to achieve cost comparable not only to the algorithm with the best fixed $\rho$, but also to the best strategy of switching between multiple values of $\rho$ a bounded number of times. This can be useful in scenarios where well-predictable parts of the input are interleaved with unpredictable or adversarial sequences.

## 5 Experiments

We ran an experimental evaluation of our algorithms compared to existing learning-augmented ski rental algorithms for the ski rental and DPM problems. Due to space limitations, only a small part of the experimental results is presented here, and others are deferred to the supplementary material, which also contains the source code and data [9]. Our results suggest that the performance of learning-augmented algorithms indeed degrades smoothly when the error increases, providing solutions which are better, for medium errors, than naive algorithms trusting the predictions and online (predictionless) algorithms. In the experiments, our algorithm's performance degrades more smoothly compared to previous learning-augmented algorithms when the prediction error increases. This is expected, since consistency-robustness trade-offs of previous algorithms optimize the two extreme scenarios of perfect predictions and adversarially bad predictions, whereas the notion of $(\rho, \mu)$-competitiveness also captures the case of useful but imperfect predictions.

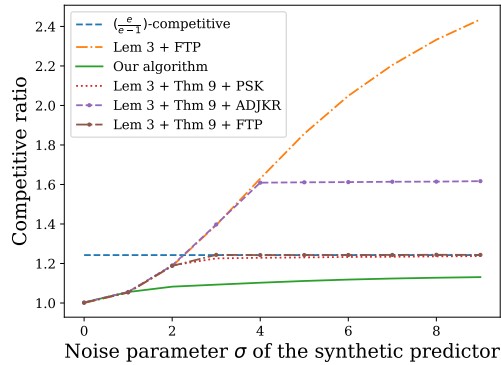
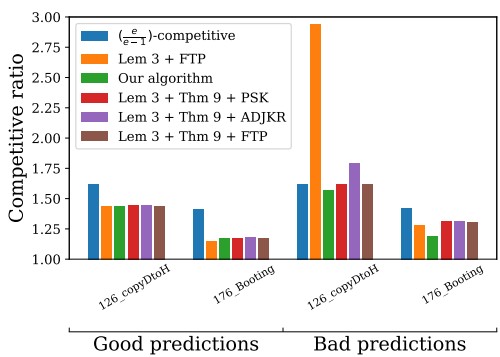

(a) Synthetic dataset with durations from $U[0, 8]$. Predictions are generated by adding Gaussian noise with standard deviation $\sigma$ to the exact durations.

(b) Real-world dataset [45] based on smartphone I/O traces. Predictions come from a multiplicative weights update algorithm [24; 23] run on past idle periods.

Figure 3: Experimental performance of different algorithms for DPM with four power states.

**Setup.** We consider four power states, whose respective power consumptions are $\{1, 0.47, 0.105, 0\}$ and wake-up costs are $\{0, 0.12, 0.33, 1\}$, values corresponding to the *active*, *idle*, *stand-by* and *sleep* states of an IBM mobile hard-drive [28]. We convert algorithms initially designed for ski rental using Lemma 3, and convert the resulting algorithms to their prudent variants as discussed in Section 3. In addition to the classical $e/(e-1)$-competitive online ski rental algorithm, we consider the following algorithms: FTP, which blindly follows the prediction (i.e., it either buys at time 0 or never); PSK, the randomized algorithm from [40]; and ADJKR, the deterministic algorithm from [4]. We use Theorem 9 to let the algorithms automatically adjust their consistency parameters over time ($\rho$ in the notation of our paper). For FTP, when we use Theorem 9 we combine it only with the classical randomized online algorithm. More details about the setup and additional scenarios are provided in the supplementary material.

**Synthetic scenario.** For the synthetic dataset, we generate both the input data and predictions similarly to Purohit et al. [40]. Figure 3a shows the performance of the algorithms on a dataset composed of $10\,000$ idle periods, whose durations are drawn independently and uniformly from $[0, 8]$. We feed the learning-augmented algorithms with synthetic predictions generated as follows: each prediction is equal to the exact request plus a random noise drawn from a normal distribution of mean 0 and standard deviation $\sigma$ (rounding any negative predictions to 0). We can observe that for low error the algorithms perform equally well. The figure shows that the algorithms PSK and ADJKR (combined with Lemma 3 and Theorem 9) essentially perform as well as the better of the two algorithms FTP and the classical online algorithm without predictions. In contrast, since our algorithm not only optimizes consistency and robustness, but also $(\rho, \mu)$-competitiveness, our algorithm achieves a significant improvement over previous algorithms in the regime of medium-sized errors and even when predictions are only very weakly correlated with the truth.

**Real-world scenario.** We created a dataset based on I/O traces from a Nexus 5 smartphone [45]. Predictions are generated based on past idle periods in a way proposed by Helmbold et al. [23] in the context of spinning down disks of mobile computers. The predictor adapts the *Share* learning algorithm of Herbster and Warmuth [24], which is based on the multiplicative weights update method. Since it is interesting to evaluate learning-augmented algorithms both in the presence of good and bad predictions, we consider two variants of that predictor. The good variant uses hyperparameters proposed in [23]; the bad variant has the rate parameter of weight updates negated. Figure 3b presents the results of the experiment on two selected traces. In particular, on each dataset, either our algorithm performs better than all the others, or the nonrobust FTP is the best one and all robust learning-augmented algorithms are almost equally good.

## Acknowledgments and Disclosure of Funding

Christian Coester is supported by the Israel Academy of Sciences and Humanities & Council for Higher Education Excellence Fellowship Program for International Postdoctoral Researchers. Research was carried out while he was at CWI in Amsterdam and supported by NWO VICI grant 639.023.812. Marek Eliáš was supported by NWO GROOT project number OCENW.GROOT.2019.015 (OPTIMAL) and research was carried out while he was at CWI in Amsterdam. Adam Polak is supported by SNSF project *Lattice Algorithms and Integer Programming* (185030).

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
