# OpenReview forum: "Learning-Augmented Dynamic Power Management with Multiple States via New Ski Rental Bounds"
_NeurIPS.cc/2021/Conference — NeurIPS 2021 Poster_

### Official Review · Reviewer_53BA · 2021-07-05

**Rating:** 6
**Confidence:** 4

**Summary:**

This paper looks at learning-augmented versions of the ski rental problem, and derives theoretical bounds (upper bounds via an algorithm and lower bounds) on the competitive ratio that can be achieved with noisy hints.  It then additionally extends these results to other variations of the problem (dynamic power management, metrical task systems);  in particular, this gives a practical motivation for this particular study.

**Limitations And Societal Impact:**

Because this is primarily a theoretical paper and I do not think we need worry about potential negative social impacts;  I do appreciate the potential positive impact of the primarily theoretical work in that the authors do describe a potential real-worldd application that is reasonably realistic. I believe the authors have addressed from the theoretical perspective the limitations of their work.



**Main Review:**

I am generally a fan of learning-augmented algorithms, and believe it is a good direction for the theoretical computer science community to go to make their work more relevant. I acknowledge that sadly it is not clear the theory community is entirely on board yet. It is perhaps less clear how the work in learning-augmented algorithms generally has an immediate effect on the AI community, although as with any application of AI, I think the application area may, in many circumstances, be of interest to the AI community.

This paper covers one of the "most natural" online algorithm problems (ski rental problem), which is well known and well studied in the online literature.  I am not clear if the ski rental problem and its variations are of wider interest to the AI community, but it is a fundamental problem, and the authors do a suitable job of providing a practical motivation (dynamic power management) that increases the "weight" of the problem being studied.  The theoretical work is well developed and would be worthy of publication in a theoretical conference, although it is perhaps unclear what level of conference it would reach.  The results are complete and in particular the characterization of the tight lower and upper bound is compelling.  The paper is well presented.

Because learning-augmented algorithms is such a recent area it is a bit difficult to determine what is the right "level" for acceptance.  I am reading a number of papers on this theme;  this is clearly one of the better ones.  I do think it provides a useful and fairly compelling example of the genre, so while I would still deem it borderline I would be inclined to accept, although I am open to the viewpoints of the other reviewers.

[Final Note;  I have considered the author response.  I believe the authors attempted to address the concerns presented in a reasonable manner.  While I am not changing the final score -- I still think it is above the acceptance threshold but only marginally so -- I remain inclined to accept the paper, which seems consistent with other reviews.]

**Time Spent Reviewing:**

1.5

---

> ### Author Response · Authors · 2021-08-10
> **Response to Official Review of Paper8547 by Reviewer 53BA**
>
> Thanks a lot for your review. On the question of relevance of ski rental to the AI community (beyond the learning-augmented perspective), there are indeed links with ML problems as well. For example, in expert learning with switching cost, the decision whether to switch to a better expert is essentially a ski-rental type decision (buying cost corresponds to switching cost and renting cost corresponds to the difference in loss between two experts). We will better highlight such connections in our paper.
>
> Please also note our answer to Reviewer da7H where we discuss additional experiments with real-world data.

---

### Official Review · Reviewer_SQBd · 2021-07-14

**Rating:** 6
**Confidence:** 4

**Summary:**

This paper considers the online version of the dynamic power management problem with multi-states (DPM). In this problem, the machine has multiple power-save states, different power-save states have different active cost and wake-up cost. Initially, the machine stays in the busy state. The input contains a set of idle periods during which the machine can stay in some power-save state. The length of each idle period is not known by the algorithm. When an idle period is released, the algorithm can make the machine stay in some power-save state for some length smaller than the length of the idle period and pay the corresponding active cost. At the end of each idle period, the algorithm needs to set the machine to be in the busy state and pay the corresponding wake-up cost. The objective is to minimize the total energy consumption of this off and on process. A new online algorithm for the ski rental problem is proposed which can achieve tight approximation ratios together with possessing a monotone property which is crucial in being used to solve the DPM problem studied in this paper. As a by product, for multi-states DPM, a randomized online algorithm with a good approximation algorithm can be derived based on the results in this paper.

**Limitations And Societal Impact:**

Nil.

**Main Review:**

The main result of the paper is a learning-augmented algorithm for multi-state DMP that achieves a good consistency and robustness at the same time (Corollary 8). The robustness directly implies a e/e-1 randomized online algorithm for general DMP. To achieve the above result, the authors give a new algorithm for the ski-rental problem which is equivalent to 2-state DMP. And then, the authors run k ski-rental algorithms in parallel in each idle period. The correctness of the above reduction between 2-state DMP and multi-state DMP relies on the fact that the algorithm for 2-state DMP has a monotonic property. The main technique in the algorithm for the ski-rental problem is a probability density function defined by the authors. By carefully choosing the value of the density function according to the predicted value, the authors show that the algorithm is optimal. Finally, the authors show another similar algorithm that almost achieves the same performance but suitable for a wider range of predictions. The main technique of this part is the reduction between DMP and  Metrical Task System.

This work is the initial study of the multi-state DMP in the learning augmented setting. The proposed algorithm achieves the optimal ratio e/e-1 when the prediction is untrusted and therefore the algorithm is robust. The competitive ratio of the proposed algorithm can be arbitrarily close to 1 when the prediction is reliable and thus the algorithm is consistent. Although there are some results that build on the previous results, the overall technique is sound and non-trivial. On the other hand, the decision about which results are theorems or corollaries seem not quite suitable in some places. For example, some theorems should be stated as a lemma, like Theorem 3, while some lemma or corollary should be stated as a theorem, like Corollary 8.

There is also a lack of high level illustration of the importance of certain results, or the comparison with earlier results. For example, for the two earlier papers about ski rental in the learning framework, do their algorithms satisfy the monotone property. If no, then the contribution of this paper is much more since they give a new algorithm not only satisfying some property which earlier algorithms do not satisfy but also has better bounds than previous algorithms.

A minor correction:

Page 5, line 182, should be A’.



**Time Spent Reviewing:**

20

---

> ### Author Response · Authors · 2021-08-10
> **Response to Official Review of Paper8547 by Reviewer SQBd**
>
> Thanks a lot for your review. In the following we try to respond to/clarify the points raised:
>
> - Regarding high-level illustration of importance and comparison with earlier results: We will expand this discussion in the final version. The ski rental algorithms in earlier papers are actually monotone as well (and they achieve this "automatically", so it may seem a bit surprising that it was non-trivial in our case. E.g. taking the algorithm that optimizes $\mu$ as a function also of $\tau$ wouldn’t work). The major differentiating factor between our and previous ski-rental algorithms is that our guarantee is a smooth (tight) function of the prediction error (which is also used in the DPM reduction), whereas previous works focused on the two extremes of perfect predictions vs worst-case predictions. Although Purohit et al. [35] actually also state a guarantee that could be phrased as $(\rho,\mu)$-competitiveness, in their case one has $\mu=\rho\ge 1$. Note that this guarantee is already trivially achieved by the algorithm that fully trusts the predictions, which is $(1,1)$-competitive. Interestingly, the best version of the algorithm of [35] in our experiments on the dataset of [35] is the $(1,1)$-competitive one. This might be additionally indicative of $(\rho,\mu)$-competitiveness being more useful for practical performance than merely optimizing robustness and consistency. (Of course, via Theorem 5 we still also get optimal robustness and consistency.) Our new experiments (see our response to Reviewer da7H) also show a significant improvement on real-world data.
> - We agree about reclassifying Theorem 3 as a lemma, and will apply this to the paper.

---

### Official Review · Reviewer_Bp21 · 2021-07-18

**Rating:** 7
**Confidence:** 3

**Summary:**

The authors provide a new online algorithm for the Dynamic Power Management problem and Ski Rental problem in the learning-augmented setting where the learner has access to predictions.  With inaccurate predictions the algorithm performs as well as classical algorithms, but the algorithm has improved competitive ratio with accurate predictions.  A lower bound is also provided and the algorithm is near optimal.

**Limitations And Societal Impact:**

Yes.

**Main Review:**

The paper deals with the important setting of dynamic power managment, which is a practically useful problem.  To do this, they produce a new ski-rental algorithm in the learning augmented setting.  The addition of predictions seems useful to make the settings more realistic.

The paper is well written and appears to be correct. The work is novel and seems technically challenging, especially the main ski-rental algorithm.  Reduction of DPM to SR seems common, but the use of the monotonicity of the CDFs for this particular ski-rental algorithm was interesting. Using online learning to provide bounds simultaneously for all $\rho$ is important to have a general algorithm, but does not seem to be a main contribution of this work. The lower bound for the learning-augmented setting also strengthens the result.  The experiments don't seem completely necessary for this work but are good to have for completeness.

**Time Spent Reviewing:**

4

---

> ### Author Response · Authors · 2021-08-10
> **Response to Official Review of Paper8547 by Reviewer Bp21**
>
> Thanks a lot for reviewing our submission and your positive comments.

---

### Official Review · Reviewer_da7H · 2021-07-23

**Rating:** 7
**Confidence:** 4

**Summary:**

The paper considers the control of passing between power-on and several stand-by or idle states for devices like CPUs that may have non-deterministic job arrivals.  The authors model this problem as a ski rental problem, where a prediction on the length of the state is available.  They assume some predicted value with varying noise to emulate predictions coming from a learning algorithm.  They propose an algorithm for this multi-state ski rental problem with 'learned' predictions and numerically compare to other known algorithms.

**Limitations And Societal Impact:**

Yes.

**Main Review:**

The paper is well written and easy to follow.

The motivation for this work was power management, so experiments that show how this algorithm performs in that real scenario would make the paper stronger.

What is the scale of noise used in the experiments, and how should we interpret them?  The noise value in the results goes up to roughly 5, but it's unclear what this means.  Further, at this high end other algorithms like ADJKR trend towards performing better than the proposed algorithms.  Some explanation and discussion here is necessary.

The authors mention 'learning augmented' algorithms.  It's not clear what learning is happening - are the predictions being refined or learned online ?  This doesn't seem to be the case.  It seems the predictions are just some noisy value around the true period lengths?

The paper ends abruptly in the Experiments section.  Some concluding remarks on the results from the experiments are missing.

**Time Spent Reviewing:**

4

---

> ### Author Response · Authors · 2021-08-10
> **Response to Official Review of Paper8547 by Reviewer da7H**
>
> Thanks a lot for your review. We believe that we can address your constructive suggestions:
>
> - We agree with your suggestion that an experiment with a real-world scenario would strengthen our paper, and so we decided to run such an additional experiment. Unfortunately, although some of the previous work on DPM that we cite includes experiments, these papers are not recent and the datasets used in them are no longer available. For that reason, we used I/O traces from Nexus 5 smartphone available at http://iotta.snia.org/traces/block-io. We took the five largest traces, from which we extracted durations of idle periods between requests. We haven't found power states specification for that smartphone, hence we used the same power states as for the synthetic experiments, i.e. an IBM mobile hard-drive's power states reported in [25]. Because of that, we had to scale up idle periods so that they have similar order of magnitude as in the synthetic experiments. For this real-world dataset, we also implemented a simple “real” predictor that generates predictions from past data using a multiplicative weight “Share update” learning algorithm of Herbster and Warmuth [21]. The competitive ratios obtained with this predictor, combined over the above smartphone datasets, are as follows:
> >
>        1.133 Our algorithm
>        1.143 FTP
>        1.145 PSK
>        1.146 Robust FTP
>        1.147 ADJKR
>        1.401 Classical Online e/(e-1)-competitive
> More detailed results for specific datasets can be seen here: https://imgur.com/a/RQ71Pmr
> In particular, on each dataset, either our algorithm performs better than all others, or blindly following predictions (FTP, non-robust) is best and our algorithm is almost as good. We will include these new real-world experiments in the final version of our paper.
> - The noise parameter $\sigma$ is the standard deviation of the Gaussian noise. For example, in Figure 3 a value of $\sigma=5$ means that the prediction is equal to the true period length (which was drawn from [0,4]) plus a random $N(0,5)$ noise. We mentioned this in the text, but it may have been easy to miss, so we will use a more descriptive text in the figure captions to make it more obvious.
> - You have a very good point about ADJKR performing better than our algorithm for high error, this was in fact a bug in our implementation of Theorem 9 (affecting the right part of Figure 3 as well as Figures 4 and 5 in the supplementary material, see corrected Figure 4 here https://imgur.com/a/wUZXEUL and the other two under the link below). For the Figures for multi-state DPM, there is an additional interesting improvement we would like to point out: A key weakness of all randomized algorithms considered in our paper is that they often assign a non-zero probability to *all* power states. In practice, as we now found in the literature, one can do a post-processing to the probability vector so that at all times there are at most two power states with non-zero probability (more precisely: If B is the expected wake-up cost given the current probability distribution over power states, then consider the alternative algorithm that assigns non-zero probability to only the two power states with wake-up cost adjacent to B, with probabilities chosen such that the expected wake-up cost is still exactly B. The expected running cost of this alternative algorithm can only be lower than for the original algorithm. Note that this post-processing can be applied online). Applying this conversion to all algorithms, we observed significant performance improvements in all of them (except ADJKR, which is deterministic and therefore unchanged under this conversion). Re-running the experiments with this improvement (and the bug fixed), the new versions of affected figures become: https://imgur.com/a/tDG6iIq
>  - You are right that in our experiments we did not apply learning to obtain the predictions. Methods to learn such predictions for DPM exist (we mention Chung et al. in the intro and give further references in related work). Since the focus of our work is to use such predictions rather than generate them, we used artificial predictions (of the same type as in Purohit et al.) in the experimental section to showcase how performance evolves as the prediction error varies. However, in our new real-world experiments mentioned above we now additionally use real learned predictions.
>  - As you suggest, we will add a discussion of the experimental results in the final version of our paper. In particular, one observation (especially from the new figures) is that our algorithm benefits from the predictions even if they are only very weakly correlated with the truth. This does not seem to be the case for previous ski rental algorithms, which aimed at optimizing the performance only in the two extreme scenarios of perfect predictions and adversarial predictions, whereas the tight guarantee of our algorithm also covers the case of useful but imperfect predictions.
>
> Please let us know if you have further questions.

---

> > ### Comment · Reviewer_da7H · 2021-09-01
> > **Thank you for the clarifications**
> >
> > I'm satisfied with your response to my suggestions.

---

### Decision · Program_Chairs · 2021-09-28

**Decision:**

Accept (Poster)

**Comment:**

The reviewers generally agree that dynamic power management is an interesting problem that motivates further study of the ski-rental problem in the learning-augmented algorithms framework. Reviewers da7H and Bp21 both say that the paper is well-written and organized. Reviewers Bp21, SQBd, and 53BA found the theory well developed and insightful (especially the reduction from DPM to SR and using the fact that it preserves monotonicity). On the other hand, reviewer SQBd asked for more high level illustrations of the important results and comparisons with prior work, which would be good to add to the paper. Reviewer da7H asked for real world experiments, and the authors provided these during the rebuttal period. If possible, it would be good to add these to the final version of the paper. Overall the paper has good support from the reviewers, so I recommend accept.

**Consistency Experiment:**

NeurIPS has a long history of experimentation. In 2014, NeurIPS ran an experiment in which 10% of submissions were reviewed by two independent committees to quantify the randomness in the review process. This year, we repeated a variant of this experiment to see how the quality of the review process has changed over time.  This paper was part of the experiment and was therefore assigned to two committees (consisting of reviewers, an Area Chair, and a Senior Area Chair) that reached independent decisions.  If both committees made the same recommendation, this recommendation was followed. If a single committee recommended acceptance, the paper was accepted (with the exception of a few cases in which the other committee identified what we considered a fatal flaw, e.g., an error in a key result).

Both committees reached the same decision: **Accept (Poster)**

The other committee assigned to the paper recommended **Accept (Poster)**.  You can find the other set of reviews, along with any follow up discussion with the authors here:
https://openreview.net/forum?id=xkQ4MhLv52X